# LEA: Learning Latent Embedding Alignment Model for fMRI Decoding and Encoding

**Xuelin Qian**[*]                                        *xuelinq92@gmail.com*
*School of Data Science*
*Fudan University*

**Yikai Wang**[*]                                         *yi-kai.wang@outlook.com*
*School of Data Science*
*Fudan University*

**Xinwei Sun**                                            *sunxinwei@fudan.edu.cn*
*School of Data Science*
*Fudan University*

**Yanwei Fu**[†]                                          *yanweifu@fudan.edu.cn*
*School of Data Science*
*Fudan University*

**Xiangyang Xue**                                         *xyxue@fudan.edu.cn*
*School of Computer Science*
*Fudan University*

**Jianfeng Feng**                                         *jianfeng64@gmail.com*
*Institute of Science and Technology for Brain-inspired Intelligence*
*Fudan University*

**Reviewed on OpenReview:** *https://openreview.net/forum?id=89QT2DsKyj*

## Abstract

The connection between brain activity and visual stimuli is crucial to understanding the human brain. Although deep generative models have shown advances in recovering brain recordings by generating images conditioned on fMRI signals, it is still challenging to generate consistent semantics. Moreover, predicting fMRI signals from visual stimuli remains a hard problem. In this paper, we introduce a unified framework that addresses both fMRI decoding and encoding. We train two latent spaces to represent and reconstruct fMRI signals and visual images, respectively. By aligning these two latent spaces, we seamlessly transform between the fMRI signal and visual stimuli. Our model, called Latent Embedding Alignment (LEA), can recover visual stimuli from fMRI signals and predict brain activity from images. LEA outperforms existing methods on multiple fMRI decoding and encoding benchmarks. It offers a comprehensive solution for modeling the relationship between fMRI signals and visual stimuli. The codes are available at `https://github.com/naiq/LEA`.

## 1 Introduction

The human brain reacts dynamically to visual stimulus from the eyes (Teng & Kravitz, 2019). We can indirectly measure these responses using functional Magnetic Resonance Imaging (fMRI). Identifying and categorizing distinct patterns of brain activity in response to visual stimuli is a crucial step in understanding

---

[*]:Equal contribution. [†]: Corresponding author.

the human brain. One method of achieving this is inverse modeling (Parthasarathy et al., 2017; Horikawa & Kamitani, 2017), reconstructing the visual stimulus from the fMRI signal. Since the brain may not capture all the details in an image, an exact reconstruction is not necessary. Researchers are mainly focused on decoding the semantics of images (Horikawa & Kamitani, 2017; Shen et al., 2019; Beliy et al., 2019; Gaziv et al., 2022). On the other hand, fMRI encoding tries to predict the fMRI signal from visual stimulus. It is important to note that semantic information in fMRI data is spread out and closely linked between nearby elements (Chang et al., 2019). In real-world situations, there are not always many pairs of images and signals available. This makes it difficult to use deep learning methods effectively, as they often require large datasets to understand fMRI data.

Recent researches (Ozcelik et al., 2022; Ferrante et al., 2022; Chen et al., 2022; Liu et al., 2023; Du et al., 2023; Ozcelik & VanRullen, 2023) decode fMRI signals based on pre-trained generative models such as Instance-Conditional GAN (Casanova et al., 2021), diffusion models (Ho et al., 2020), masked autoencoders (He et al., 2022), CLIP (Radford et al., 2021), to name a few. Despite achieving impressive results in high-fidelity generation, they face several challenges: (1) Despite their ability to generate images of exceptional quality, pre-trained generative models struggle to ensure semantic consistency with fMRI signals, presenting an ongoing challenge. (2) These models have demonstrated the capability to produce high-quality images even when provided with random noise disguised as fake fMRI signals (see Sec. 4 for details). This raises concerns about their reliability, particularly when dealing with a wide range of visual stimuli, including those not seen during training (out-of-distribution). This reliability issue is particularly relevant to fMRI decoding. Thus when utilizing large-scale pre-trained generative models for fMRI decoding, distinguishing between genuine fMRI signal decoding and unconditional image generation becomes a critical concern.

This paper proposes a novel approach to address the challenges associated with fMRI signals by simultaneously tackling the tasks of fMRI decoding and encoding. Our approach designs encoder-decoder architectures specifically for both fMRI signals and images, avoiding the need for paired image-fMRI data. These encoder-decoder structures effectively learn dense and compact latent representation spaces. As a result, our model achieves the best performance on fMRI zero-shot classification where the categories of fMRI signals are unseen during training. Moreover, these encoder-decoder architectures can be trained via self-supervised learning techniques (Vincent et al., 2008; He et al., 2020; 2022), eliminating the need for additional supervision. Particularly, we introduce the ROI embedding layer to tackle the individual mismatch of fMRI signals, enabling the pre-train on multi-subjects without manual alignment of different subjects. Furthermore, this learned representation serves to reduce the feature dimension of both modalities, thus simplifying the task of transformation between them.

Technically, to align the two learned representation spaces effectively, we propose a lightweight alignment module that can be trained with a relatively small set of fMRI-image pairs. Alignment across diverse types of signals has been extensively explored in various vision-related tasks, such as image-text (Radford et al., 2021; Jia et al., 2021; Mahajan et al., 2018) and image-audio (Arandjelovic & Zisserman, 2017; Morgado et al., 2021; Owens & Efros, 2018; Patrick et al., 2020). These studies have revealed that well-encoded signals from different modalities can be effectively aligned within a shared representation space. However, given the absence of a large-scale fMRI-image paired dataset to support end-to-end training of such an alignment model, inspired by the linear relationships observed between text- and vision-only models (Merullo et al., 2023), we employ a linear regression model to learn the mapping between the two latent spaces. Furthermore, this lightweight linear model can be trained with a small number of fMRI-image pairs, eliminating the need for a large-scale paired dataset and enabling the creation of individual-specific models. Within the framework of LEA, the alignment module serves a crucial role in assessing the reliability of both fMRI encoding and decoding models through image-fMRI-image reconstruction. By initially estimating fMRI signals from images and subsequently reconstructing the visual stimuli, the ability to generate semantically consistent images serves as a reliability indicator.

Consequently, we have devised the Latent Embedding Alignment (LEA) framework for fMRI decoding and encoding. LEA encompasses two distinct encoder-decoder architectures for handling fMRI signals and images, respectively. We introduce an alignment module to facilitate the transformation between the latent representation spaces of fMRI signals and images. As illustrated in Figure 1, when reconstructing visual stimuli from fMRI signals, we encode the fMRI signal, convert it into a latent image embedding, and

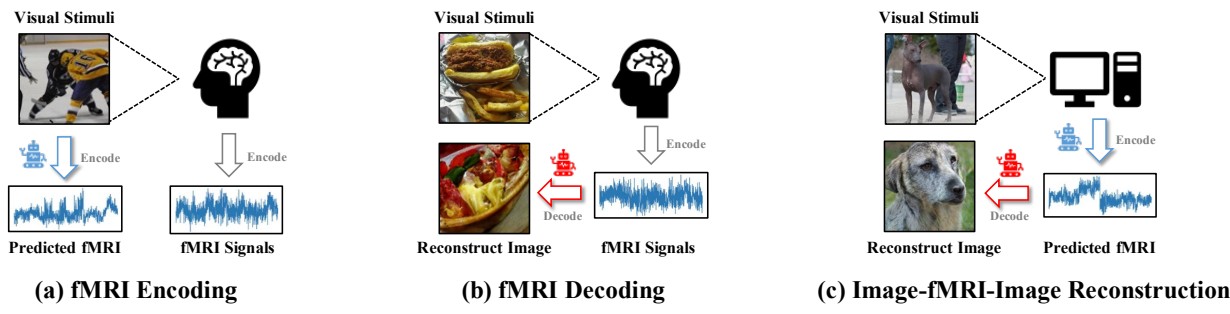

**(a) fMRI Encoding**      **(b) fMRI Decoding**      **(c) Image-fMRI-Image Reconstruction**

Figure 1: Our proposed LEA efficiently enables (a) fMRI encoding to estimate the brain activity from visual stimuli; (b) fMRI decoding to recover the visual stimuli from recorded fMRI signals, and (c) image-fMRI-image reconstruction to ensure the reliability of fMRI decoding and encoding.

subsequently decode it to generate images. Conversely, when predicting brain activity, we encode the images, transform them into latent fMRI embeddings, and then decode them to produce fMRI signals. An important feature of LEA is its support for image-fMRI-image reconstruction, which serves as an indicator for the reliability of both fMRI encoding and decoding models. Through extensive experimentation on BOLD5000 and GOD benchmarks, we demonstrate that LEA exhibits both efficiency and effectiveness in fMRI decoding and encoding.

**Contribution**. Our key contributions can be summarized as follows:

- **Encoder-Decoder Architectures**: We introduce specialized encoder-decoder architectures that learn latent representations for fMRI signals and images without the need for extensive paired training data.
- **ROI Embedding Layer**: We devise an innovative ROI embedding layer to extract fMRI signals, ensuring geometric consistency in feature extraction. Additionally, we propose the use of global representations for fMRI signal reconstruction, promoting efficient feature compression.
- **Latent Space Alignment**: We establish a linear connection between the latent spaces of fMRI and images, enabling seamless fMRI decoding and encoding within a unified framework called LEA.
- **Superior Performance**: Through rigorous testing on multiple benchmark datasets, we demonstrate the exceptional performance of LEA, underscoring its effectiveness and significant advancements in the field of fMRI decoding and encoding.

## 2 Related Work

**fMRI Encoding and Decoding.** fMRI encoding involves learning the mapping from visual stimuli to fMRI signals. Early studies (Yamins et al., 2013; 2014; Güçlü & van Gerven, 2015; Yamins & DiCarlo, 2016) primarily employed convolutional neural networks to extract semantic information from visual stimuli. However, with the resurgence of neural language models, recent research has successfully utilized models such as BERT (Devlin et al., 2018), Transformer (Vaswani et al., 2017), and GPT-2 (Radford et al., 2019) to predict fMRI responses from stimuli, including images and even words/sentences. Conversely, fMRI decoding focuses on reverse mapping, from brain activity to stimuli. Previous studies (Mozafari et al., 2020; Ozcelik et al., 2022; Chen et al., 2015; 2016; Zhang et al., 2016) developed regression models specifically designed to extract valuable information from fMRI signals. Ozcelik et al. (2022) employed a pre-trained Instance-Conditional GAN model (Casanova et al., 2021) to reconstruct images by decoding latent variables from fMRI data. Ferrante et al. (2022) utilized pre-trained latent diffusion models (Ho et al., 2020) to generate stimulus images by mapping fMRI signals to visual features. Beliy et al. (2019) adopted self-supervised learning on testing data to enable adaptation to testing statistics. Chen et al. (2022) introduced a self-supervised sparse masked modeling strategy to encode fMRI data into latent embeddings and fine-tuned latent diffusion models with double conditioning. Huo et al. (2024) further ensures that the decoded images retain both the semantics and stucture. In this paper, we present a unified framework that efficiently realizes both fMRI encoding and decoding.

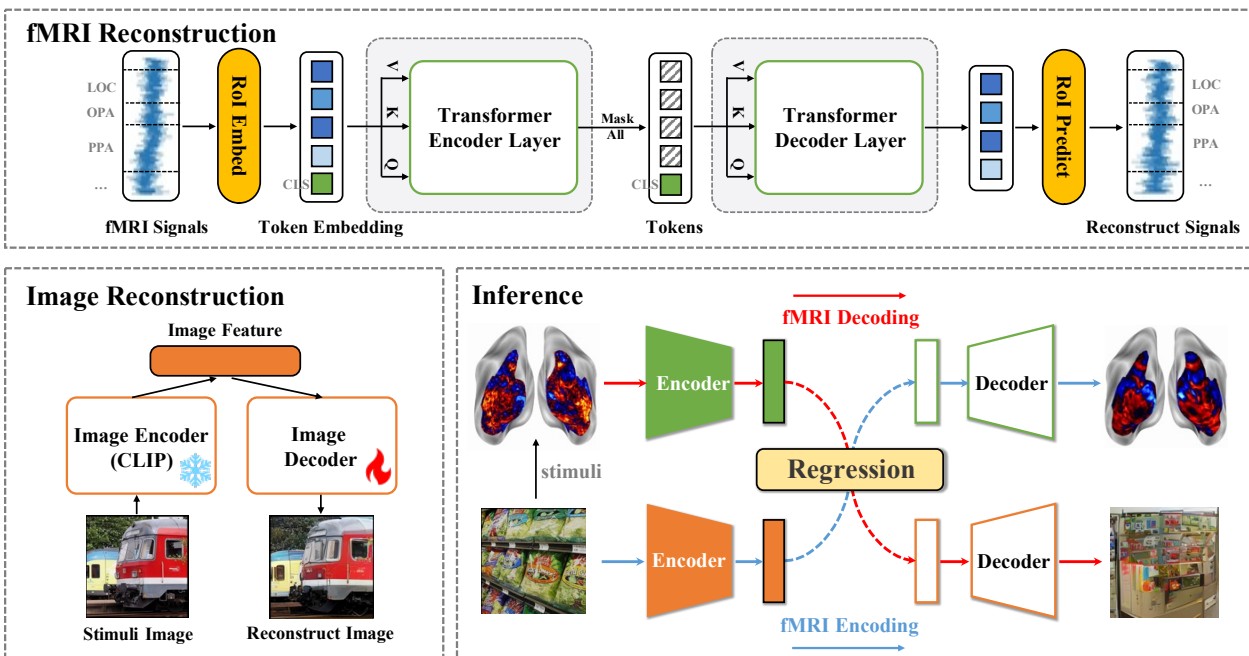

Figure 2: The overview of our proposed LEA framework. We learn a latent representation space via the encoder-decoder architectures designed specifically for fMRI signals and images. Then a lightweight alignment module is proposed to link the two latent spaces, enabling both fMRI encoding and decoding.

**Multi-Modality Alignment.** It aims to establish connections across different modalities, such as image-text alignment (Radford et al., 2021; Jia et al., 2021; Mahajan et al., 2018) and image-audio alignment (Arandjelovic & Zisserman, 2017; Morgado et al., 2021; Owens & Efros, 2018; Patrick et al., 2020). Girdhar et al. (2023) endeavors to align multiple modalities within a unified latent representation space. The majority of efforts in this direction are centered around the CLIP model (Radford et al., 2021), which is trained on extensive image-text pairs to learn a shared latent space minimizing the distances between correlated image-text pairs. For instance, Liu et al. (2023) bridged the modality gap by leveraging CLIP's cross-modal generalization ability to address the limitation of limited fMRI-image paired data. Similarly, Du et al. (2023) employed multi-modal deep generative models to capture the relationships between brain, visual, and linguistic features. Ozcelik & VanRullen (2023) predicted multi-modal features from fMRI signals and generated reconstructed images using a latent diffusion model (Rombach et al., 2022).

## 3 Methodology

### 3.1 Problem Setup

Given a fMRI signal $\mathbf{f} \in \mathbb{R}^L$ recorded from brain activity and the corresponding visual stimuli $\mathbf{I} \in \mathbb{R}^{H \times W \times 3}$, the purpose of our work is to learn a unified framework that can perform both **fMRI decoding** task to recover the observed image from fMRI signal and **fMRI encoding** task to predict the brain activity from the image.

For the decoding task, the ideal scenario involves the reconstructed images $\hat{\mathbf{I}}$ matching the real ones precisely. However, the unique nature of biological mechanisms suggests that individual memory or attention can influence an individual's brain activity response to the same image, and thus variations may occur. In light of this, we adhere to the approach introduced in (Mozafari et al., 2020; Ozcelik et al., 2022), requiring the recovered images $\hat{\mathbf{I}}$ to exhibit semantic consistency with $\mathbf{I}$. Consequently, our primary focus is on the *semantic fMRI decoding* task.

For the encoding task, it is challenging to make precise predictions of element-wise fMRI signals due to their inherent noise and redundancy. Therefore, in this paper, our focus shifts towards predicting the overall trends in fMRI signals. More specifically, we aim to generate estimated fMRI signals that exhibit a correlation with the actual fMRI signals. This correlation serves as an indicator of whether a Region-Of-Interest (ROI) region is activated or not. To achieve this goal, our objective is to produce estimated fMRI signals based on the visual stimulus, that is highly correlated with the actual fMRI signals.

## 3.2 Latent Space Construction for fMRI

The fMRI signals indirectly record neural activity in the visual cortex of the brain by measuring the blood-oxygen-level-dependent (BOLD) signals, which are usually represented as data of 3D voxels, covering visual regions, like early visual cortex (EVC), lateral occipital complex (LOC), occipital place area (OPA), parahippocampal place area (PPA), and retrosplenial complex (RSC) (Horikawa & Kamitani, 2015). Biological principles suggest that adjacent voxels in the brain's visual cortex often exhibit similar intensities in their stimulus-response (Ugurbil et al., 2013), resulting in redundancy within fMRI signals.

To investigate potential response patterns to visual stimuli and establish a reverse mapping for semantic encoding, it is essential to focus on the relationships among different regions while simultaneously capturing activity changes across long-range voxels. Inspired by the advantages of long-range receptive fields found in transformer layers, we follow Chen et al. (2022) by adopting masked autoencoder (He et al., 2022) as the encoder-decoder model for fMRI signals.

**ROI embedding layer**. Unlike Chen et al. (2022) which equally divides the fMRI signals into patches of equal size, we maintain the voxel structure with ROI regions. Concretely, we partition the fMRI signal vector into distinct ROI regions, each potentially having different dimensions. To address the dimension inconsistency, we propose the Region of Interest (ROI) Embedding layer. Given a specific ROI region $i$, denote the corresponding fMRI signal is of size $N_i \times 1$, indicating $N_i$ voxels and 1 dimension of BOLD response, this fMRI signal region is first passed through a convolution layer with 32 kernels to extract multi-head features, resulting in dimensions of $N_i \times 32$. Subsequently, we employ a fully-connected layer to project the fMRI signal, which may have varying lengths, into a unified dimension of 1024. We then concatenate the embeddings from all ROI regions, yielding the final input of size 32M×1024, where M indicates the number of ROIs. Finally, we append a learnable [CLS] token to represent the "class" of the input.

**Global feature extraction**. In our encoder-decoder architecture training, we employ a masked auto-encoder (MAE) (He et al., 2022). The original MAE employs patch-based reconstruction, where a portion of the patches is masked, and the task is to predict these masked patches. While this approach is effective for learning local representations, it presents challenges when we aim to acquire a global latent code that represents the entire fMRI signal. To address this, we only preserve the global [CLS] token of the encoded fMRI signals, mask all the patch-based latent code, and let the decoder learn to reconstruct fMRI signals from the learned [CLS] token. This strategy proves advantageous for learning a dense fMRI representation within the [CLS] token, and the trained decoder can subsequently be used to generate fMRI signals based on the estimated latent fMRI code.

Finally, the decoder's output undergoes post-processing through the ROI Project layer. This layer mirrors the structure of the ROI Embed layer, effectively reversing the pre-processing applied to the fMRI signals, ensuring that the length matches that of the original fMRI signals. The overall model is trained by minimizing the L2 reconstruction loss of the fMRI signals.

$$\mathcal{L}_{fmri} = \frac{1}{K} \sum_{i=1}^{K} ||\mathbf{f}_i - \hat{\mathbf{f}}_i||_2^2, \tag{1}$$

where $\mathbf{f}_i$ and $\hat{\mathbf{f}}_i$ indicate the ground-truth and reconstructed fMRI signals, individually; $K$ means the totally amount of training fMRI data.

### 3.3 Latent Space Construction for Image

The image-based latent space construction is a well-established domain of research. To recover semantic content from fMRI signals, we leverage the pre-trained CLIP visual encoder as our foundation for image latent space. CLIP benefits from extensive training on large-scale image-text paired data, making its latent space rich in semantics and benefits the semantic consistency in our task.

Note that CLIP operates as an encoder-only model, and we cannot directly generate images from the image latent space. To address this issue, we adopt MaskGIT (Chang et al., 2022) as our decoder to reconstruct images from the image latent space. In this approach, we fix the CLIP visual encoder and fine-tune the class-conditional MaskGIT as a latent-conditional generative decoder. Further details regarding the objective functions can be found in (Chang et al., 2022).

### 3.4 Latent Embedding Alignment

The alignment of fMRI and image latent spaces represents a critical step in enabling the simultaneous execution of fMRI decoding and encoding tasks. Unlike conventional methods (Du et al., 2023; Liu et al., 2023) that rely on intermediary modalities to bridge the gap between fMRI and image data, our approach directly connects the latent feature representations of both domains through a linear model. This strategy leverages the expressive power of these well-defined latent spaces, eliminating the need for additional data modalities.

To address the inherent variability in fMRI signals and the potential for individual-specific patterns in latent fMRI embeddings, we employ a personalized modeling approach. Individual-specific linear models are trained to map fMRI embeddings to image embeddings and vice versa. L2 regularization, implemented via ridge regression, is used to stabilize the learning process and constrain the solution space. With the aligned latent spaces and the embedding alignment module in place, we integrate fMRI decoding and encoding tasks into a unified framework.

Specifically, the latent embedding of the fMRI is the trained [CLS] token of our fMRI encoder output, denoted as $e_{\text{fMRI}}$. The latent embedding of the image is the global visual CLIP feature, denoted as $e_{\text{image}}$. For a specific individual, we gather all training data to form two embedding matrices, $E_{\text{fMRI}}$ and $E_{\text{image}}$. Then we learn two linear mappings via the ridge regression such that:

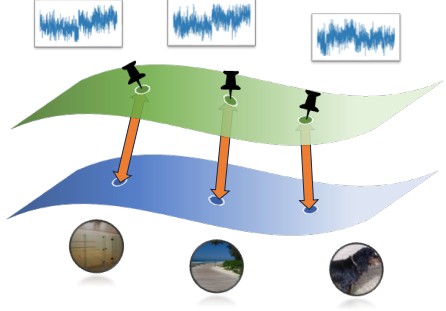

Figure 3: LEA aligns the latent space of fMRI signals and image features, enabling the bi-directional transformation.

$$e_{\text{image}} = e_{\text{fMRI}}\beta_{\text{fMRI}\to\text{image}}, \quad e_{\text{fMRI}} = e_{\text{image}}\beta_{\text{image}\to\text{fMRI}}. \quad (2)$$

The solutions are $\hat{\beta}_{\text{fMRI}\to\text{image}} = (E_{\text{fMRI}}^\top E_{\text{fMRI}} + \lambda I)^{-1} E_{\text{fMRI}}^\top E_{\text{image}}$ and $\hat{\beta}_{\text{image}\to\text{fMRI}} = (E_{\text{image}}^\top E_{\text{image}} + \lambda I)^{-1} E_{\text{image}}^\top E_{\text{fMRI}}$, respectively. We use the same regularization parameter $\lambda$ for both regression models.

In the decoding task, the fMRI signal is first encoded into the latent fMRI space. This latent representation is then projected into the latent visual feature space using the individual-specific linear model. The projected visual feature is decoded by a fine-tuned MaskGIT model to reconstruct the original visual stimuli. Conversely, in the encoding task, CLIP visual features are encoded and projected into the latent fMRI space through another linear model. The resulting latent fMRI representation is subsequently decoded to approximate the original fMRI signals.

## 4 Experiments

### 4.1 Experiment Setup

**Datasets.** We validate our LEA on Brain, Object, Landscape Dataset (BOLD5000) (Chang et al., 2019) and Generic Object Decoding Dataset (GOD) (Horikawa & Kamitani, 2017). (1) BOLD5000 encompasses

a substantial fMRI-image dataset with a total of 5254 fMRI-image stimulus trials, involving four subjects. This dataset includes 1000, 2000, and 1916 images sourced from the Scene, COCO, and ImageNet datasets, respectively. Following the protocol in (Chen et al., 2022), we choose $4,803$ images presented on a single trial for training, and the remaining 113 images for testing. We use the pre-defined five ROI regions (*e.g.*, EV, LOC, OPA, PPA, RSC) for our ROI Embed layer. (2) The GOD comprises data from 5 subjects, each viewing 1,250 images spanning 200 categories, resulting in a total of 1,250 fMRI-image pairs. We split the training set with 1200 pairs from 150 categories, and the other non-overlapping 50 classes are used for testing. The visual cortical ROI regions include V1, V2, V3, V4, FFA, PPA, LOC, and HVC.

**Metrics.** For the fMRI decoding task, we utilize the following metrics for comprehensive comparisons: N-way Classification Accuracy (Acc) (Gaziv et al., 2022) to assess the semantic correctness of generated samples, Fréchet Inception Distance (FID) (Heusel et al., 2017) to evaluate the quality of generated images, and CLIP correlation and distance to gauge the semantic consistency of generated images. For Acc we calculate the average top-1 accuracy over 1,000 trials. For FID we use 64-dimensional features extracted by the Inception-v3 model to measure their similarity to ground-truth testing images. The CLIP distance means the cosine similarity of two images and we average the similarity among all fMRI-image pairs. Meanwhile, CLIP correlation calculates the correspondence of positive fMRI-image pairs compared to others, *i.e.*, 2-way comparison of CLIP features via pair-wise correlation. For the fMRI encoding task, we employ Pearson's metric to assess the similarity between estimated fMRI signals and the ground truth. We calculate the Pearson correlation ($\times 10^2$) on each vertex of the $N \times L$ matrix, where $N$ represents the number of fMRI signals and $L$ is the dimension, and then average these correlations across different vertices.

**Setting**. We follow the general principle of not using any testing data in training. We denote our algorithm in this setting as LEA. Additionally, for a fair comparison with competitors, we follow the previous work (Chen et al., 2022; Beliy et al., 2019) and use the fMRI test set without revealing the label information. Specifically, the fMRI testing data is only trained for the reconstruction task in a self-supervised manner. When learning the latent embedding alignment module, we do not access any paired fMRI-image testing data, and the parameters of both fMRI and image reconstruction modules are frozen. These experiments are distinguished by adding the suffix of "relax" to the model name, as utilized in SSfMRI2IM (Beliy et al., 2019) and MinD-Vis (Chen et al., 2022).

**Implementation.** LEA is implemented using PyTorch. The fMRI reconstruction model is initialized with pre-trained weights from HCP datasets (Chen et al., 2022), but we randomly initialize parameters that do not match the dimensions of those from pre-trained weights (*e.g.*, ROI Embed layer). We further fine-tune it for each individual. The encoder and decoder have a depth of 24/8 with dimensions of 1024/512, and there are 16 multi-heads. For fine-tuning both models, we use the AdamW optimizer with hyper-parameters $\beta_1 = 0.9$, $\beta_1 = 0.95$, and a batch size of 8. The initial learning rate is set to 5e-5 with a weight decay of 0.01. We apply a linear learning rate schedule, gradually reducing the learning rate until it reaches the minimum value. The total number of training iterations is 100K and 300K for the fMRI and image reconstruction models, respectively.

## 4.2 Quantitative Results

**Analysis of zero-shot classification**. As we inherit the latent space from the original CLIP model, LEA enables open-vocabulary recognition of fMRI signals. To assess this capability, we perform experiments on zero-shot fMRI signal classification tasks to identify the most similar category for each fMRI signal. Specifically, we create a set of candidate class names using various text templates. These class names are then encoded using the CLIP text encoder. Subsequently, we calculate the similarity between the class embeddings and the fMRI latent embeddings learned by LEA.

We conduct experiments on GOD dataset, which contains 50 visual stimuli from 50 categories that are unseen during training. We report the top-1 classification accuracy (with the chance levels of 2%) of several competitors in Table 1. Results of competitors are from Liu et al. (2023). We divide the competitors into two groups based on their supervision modality, where "V" indicates visual supervision only and "V&T" indicates both visual and textual supervision. Our LEA trained with visual supervision beats all competitors, indicating the superiority of our bi-directional transformation.

Table 1: Top-1 accuracy of zero-shot classification on the GOD dataset.

| METHOD | MODALITY | SBJ-1 | SBJ-2 | SBJ-3 | SBJ-4 | SBJ-5 | AVERAGE |
|---|---|---|---|---|---|---|---|
| CADA-VAE (Schonfeld et al., 2019) | V&T | 6.31 | 6.45 | 17.74 | 12.17 | 7.45 | 10.02 |
| MVAE (Wu & Goodman, 2018) | V&T | 5.77 | 5.40 | 17.11 | 14.02 | 7.89 | 10.04 |
| MMVAE (Shi et al., 2019) | V&T | 6.63 | 6.60 | 22.11 | 14.54 | 8.53 | 11.68 |
| MoPoE-VAE (Sutter et al., 2021) | V&T | 8.54 | 8.34 | 22.68 | 14.57 | 10.45 | 12.92 |
| BrainCLIP-Linear (Liu et al., 2023) | V&T | 10.00 | 12.00 | 18.00 | 12.00 | 12.00 | 12.80 |
| CADA-VAE (Schonfeld et al., 2019) | V | 5.66 | 6.01 | 16.51 | 9.17 | 6.01 | 8.67 |
| MVAE (Wu & Goodman, 2018) | V | 5.30 | 5.21 | 14.13 | 8.03 | 5.44 | 7.62 |
| MMVAE (Shi et al., 2019) | V | 5.41 | 5.39 | 13.76 | 10.62 | 5.02 | 8.04 |
| MoPoE-VAE (Sutter et al., 2021) | V | 5.20 | 7.42 | 14.05 | 9.25 | 6.37 | 8.46 |
| BraVL (Du et al., 2023) | V | 8.91 | 8.51 | **18.17** | 14.20 | 11.02 | 12.16 |
| BrainCLIP (Liu et al., 2023) | V | 6.00 | 16.00 | 16.00 | 14.00 | **14.00** | 13.20 |
| LEA | V | **10.00** | **18.00** | 12.00 | **16.00** | 12.00 | **13.60** |

Table 2: The comparison of fMRI encoding with other competitors. All methods use pre-trained CLIP (Radford et al., 2021) to extract image features. The metric is the Pearson correlation coefficient ($\times 10^2$).

| METHOD | GOD | | | | | | BOLD5000 | | | | |
|---|---|---|---|---|---|---|---|---|---|---|---|
| | Sbj-1 | Sbj-2 | Sbj-3 | Sbj-4 | Sbj-5 | Avg. | CSI-1 | CSI-2 | CSI-3 | CSI-4 | Avg. |
| MinD-Vis (Chen et al., 2022) | 5.21 | 12.05 | 15.47 | 10.63 | 10.55 | 10.78 | 30.58 | 19.78 | 16.48 | 19.96 | 21.70 |
| Regressor (Gifford et al., 2023) | 6.65 | 16.22 | 18.58 | 12.82 | 14.38 | 13.73 | 32.07 | 20.58 | 20.46 | 21.18 | 23.57 |
| LEA (*relax*) | 8.43 | 15.51 | **21.20** | **19.08** | 15.07 | 15.86 | 34.32 | **26.74** | 27.74 | **24.44** | 28.19 |
| LEA | **8.49** | **17.01** | 20.58 | 17.82 | **16.86** | **16.15** | **34.78** | 25.91 | **27.86** | 24.37 | **28.80** |

**Analysis of fMRI encoding**. With two latent spaces for fMRI signals and images, LEA can not only recover the visual stimuli but also predict the fMRI signal by the visual image. As in Table 2, we conduct experiments to validate the estimation capability of our LEA. We construct two competitors to show the effectiveness of LEA. For the "Regressor", we apply liner regression to directly generate fMRI signals based on the image feature from the CLIP model. With the absence of learned fMRI embedding in LEA, the "Regressor" adopts the pipeline of 'image features → fMRI signals' instead of 'image features → fMRI features → fMRI signals'. However, such a method causes misalignment of features during reconstruction without the fMRI latent space, as validated in the empirical results. The "MinD-Vis" utilizes the transformer decoder to predict fMRI signals from CLIP image features. Such a solution is too rough and not conducive to fine-grained feature alignment and generation. Our LEA achieves the best Pearson's score, which indicates that the alignment between latent embedding spaces is more effective for fMRI prediction.

**Analysis of fMRI decoding.** We present a comprehensive comparison of fMRI decoding performance on the GOD and BOLD5000 datasets in Table 3 and Table 4, respectively. To ensure a fair assessment, we reproduce the results of competitors using their officially released models. It is worth noting that due to the availability of the official model of CIS-1,4 on the BOLD5000 dataset in MinD-Vis, we exclusively report the reproduced CLIP scores for these two individuals. Our novel LEA consistently demonstrates superior performance across various perspectives, including image quality, semantic correctness, and correlation, on both the GOD and BOLD5000 datasets. Notably, LEA excels in generating images of higher quality than its competitors on the GOD dataset. While MinD-Vis may outperform LEA on specific subjects, it significantly lags behind LEA on others, such as SUB-1,2,4 in the GOD and CSI-2,4 in BOLD5000, resulting in an overall inferior average performance. In summary, LEA not only outperforms current competitors in fMRI decoding but also possesses the capability for fMRI encoding, making it a comprehensive solution in the field.

Table 3: fMRI decoding performance comparison on the GOD dataset.

| METRICS | METHODS | SBJ-1 | SBJ-2 | SBJ-3 | SBJ-4 | SBJ-5 | AVERAGE |
|---|---|---|---|---|---|---|---|
| FID ↓ | Gaziv et al. (2022) | 7.67 | 2.67 | 2.51 | 8.93 | 2.77 | 4.91 |
| | Ozcelik et al. (2022) | 5.22 | 3.72 | 3.65 | 4.42 | 3.69 | 4.14 |
| | Vodrahalli et al. (2018) | 3.61 | 2.77 | 2.34 | 2.95 | 2.55 | 2.84 |
| | MinD-Vis (Chen et al., 2022) | 1.97 | 1.63 | 1.68 | 1.77 | 2.33 | 1.88 |
| | LEA (*relax*) | **1.45** | **1.43** | **1.52** | **1.48** | **1.25** | **1.23** |
| | LEA | 2.57 | 1.51 | 1.67 | 1.71 | 1.34 | 1.76 |
| CLIP Corr. ↑ | Gaziv et al. (2022) | 58.57 | 62.00 | 67.59 | 62.89 | 61.14 | 62.44 |
| | Ozcelik et al. (2022) | 74.02 | 80.83 | 77.65 | 75.44 | 76.46 | 76.88 |
| | Vodrahalli et al. (2018) | 69.55 | 76.38 | 79.57 | 75.34 | 75.48 | 75.26 |
| | MinD-Vis (Chen et al., 2022) | 66.69 | 77.02 | **83.78** | 74.78 | **80.04** | 76.46 |
| | LEA (*relax*) | **73.96** | 78.20 | 80.94 | **78.73** | 70.86 | 76.54 |
| | LEA | 73.79 | **79.63** | 79.96 | 76.69 | 76.32 | **77.28** |
| CLIP Dist. ↑ | Gaziv et al. (2022) | 0.31 | 0.34 | 0.34 | 0.32 | 0.32 | 0.33 |
| | Ozcelik et al. (2022) | 0.39 | 0.40 | 0.42 | 0.39 | 0.41 | **0.40** |
| | Vodrahalli et al. (2018) | 0.35 | 0.39 | 0.41 | 0.39 | 0.40 | 0.39 |
| | MinD-Vis (Chen et al., 2022) | 0.33 | 0.35 | **0.43** | 0.38 | 0.39 | 0.38 |
| | LEA (*relax*) | **0.38** | 0.40 | 0.41 | **0.40** | 0.37 | 0.39 |
| | LEA | **0.38** | **0.41** | 0.40 | 0.39 | **0.41** | **0.40** |
| Accuracy ↑ | Gaziv et al. (2022) | 1.88 | 3.79 | 12.99 | 8.45 | 6.26 | 6.67 |
| | Ozcelik et al. (2022) | 10.71 | 15.95 | 18.79 | 14.52 | 12.31 | 14.46 |
| | Vodrahalli et al. (2018) | 9.51 | 16.48 | 18.79 | 13.35 | 13.28 | 14.28 |
| | MinD-Vis (Chen et al., 2022) | 9.10 | 15.91 | **27.44** | 15.81 | 14.28 | 16.51 |
| | LEA (*relax*) | 11.18 | **18.62** | 20.45 | **20.04** | 13.15 | **16.69** |
| | LEA | **11.35** | 18.55 | 18.92 | 17.52 | **14.45** | 16.16 |

Table 4: fMRI decoding performance comparison on BOLD5000 dataset.

| METRICS | METHODS | CSI-1 | CSI-2 | CSI-3 | CSI-4 | AVERAGE |
|---|---|---|---|---|---|---|
| FID ↓ | MinD-Vis (Chen et al., 2022) | **1.20** | 1.90 | **1.40** | 1.32 | 1.46 |
| | LEA (*relax*) | 1.31 | **1.53** | 1.48 | **1.19** | **1.38** |
| CLIP Corr. ↑ | MinD-Vis (Chen et al., 2022) | **88.29** | - | - | 83.09 | - |
| | LEA (*relax*) | 86.60 | 83.56 | 83.82 | **84.77** | 84.69 |
| CLIP Dist. ↑ | MinD-Vis (Chen et al., 2022) | 0.41 | - | - | 0.37 | - |
| | LEA (*relax*) | **0.41** | 0.38 | 0.37 | **0.39** | 0.39 |
| Accuracy ↑ | MinD-Vis (Chen et al., 2022) | **29.94** | 18.50 | **21.00** | 20.37 | 22.45 |
| | LEA (*relax*) | 28.48 | **19.74** | 19.65 | **22.48** | **22.59** |

## 4.3 Qualitative Results

To intuitively illustrate the efficacy of our method, we show some generated images in Figure 4 and Figure 5 for the GOD and BOLD5000 datasets, respectively. We compare with MinD-Vis (Chen et al., 2022) and Gaziv et al. (2022) on the GOD and MinD-Vis on BOLD5000.

More concretely, our generated images consistently maintain the same semantic information as the ground truth, encompassing humans, objects, animals, architecture, and landscapes. In contrast, competitors may falter in some instances, failing to preserve semantic consistency. For instance, in Figure 4, LEA excels in producing semantic-consistent and high-fidelity images, whereas competitors struggle to preserve semantics. In the last row on the left side of Figure 5, MinD-Vis can only generate a person while neglecting the water,

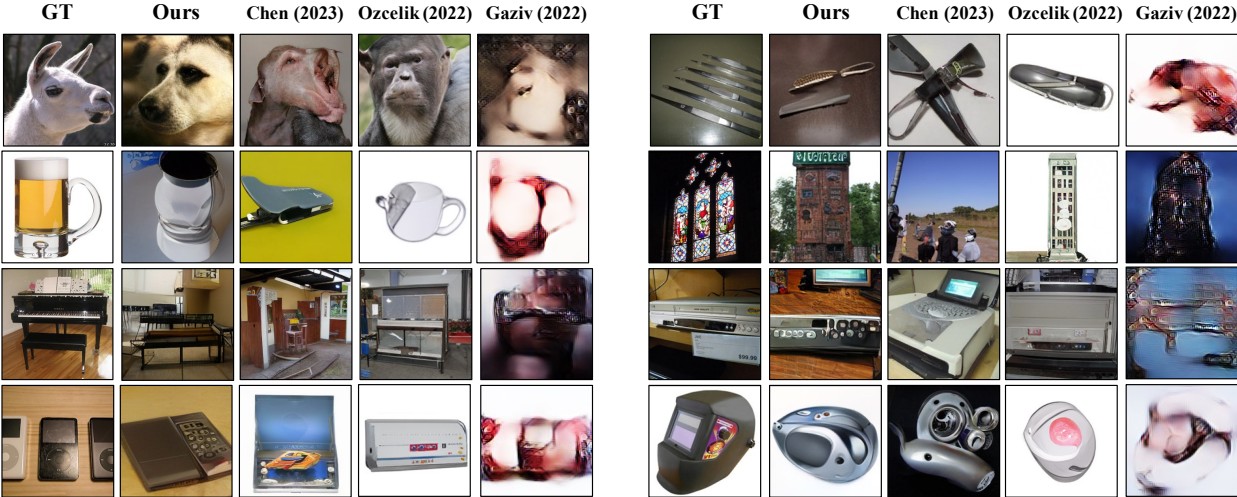

Figure 4: fMRI decoding performance comparison on the GOD dataset.

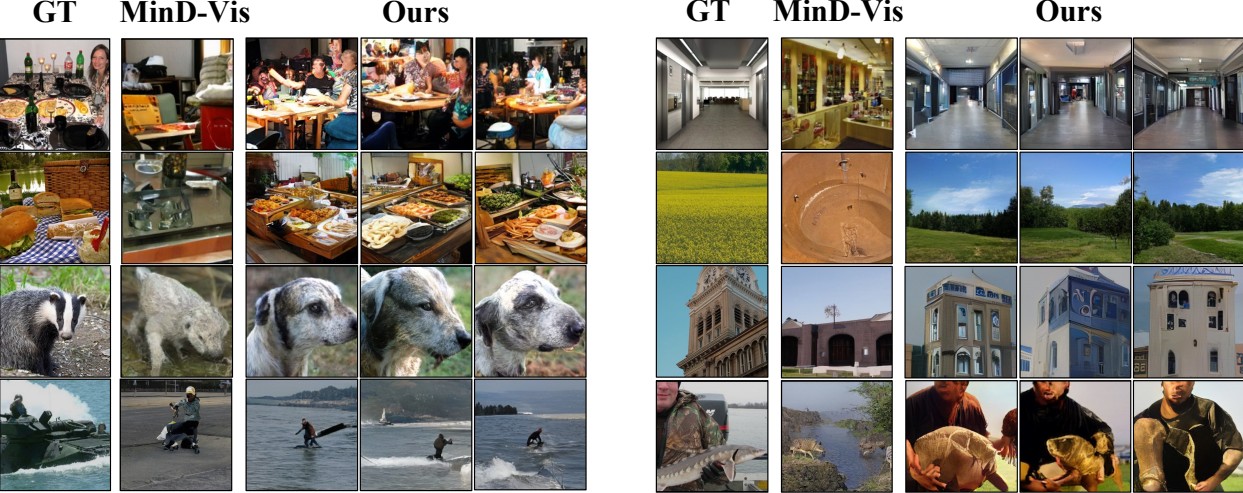

Figure 5: fMRI decoding performance comparison on BOLD5000 dataset.

whereas LEA consistently generates both the person and the water. These results highlight our ability to not only maintain the correctness of objects but also capture finer-grained semantics. Moreover, our approach achieves image fidelity in numerous cases, contributing to a more comprehensive and effective solution.

## 4.4 Ablation Studies

**Analysis of ROI Embedding Layer**. Human brains are known to be spatially segregated, and different ROI regions react to different visual stimuli. Preserving the spatial structure is beneficial to help understand the fMRI signals. Therefore, we design a ROI embedding layer ("ROIEmbed" for short) to separately encode signals from different ROI regions. Differently, previous methods (Chen et al., 2022) simply divide fMRI signals into $n = 16$ equal parts to obtain sequence features of appropriate length. Zero padding is added if the original length of fMRI signals is not a multiple of 16. We denote such a method as "PatchEmbed". To evaluate the effectiveness of our design, we conduct experiments by predicting fMRI signals from visual images and generating stimuli images from fMRI signals. In Table 5 and Table 6, our proposed ROIEmbed achieves better performance than PatchEmbed, which demonstrates its efficacy.

For the ROIEmbed layer, we extract multi-head features for each ROI region. Here, the number of multi-head is a hyper-parameter. Intuitively, if the number of heads is too small (*i.e.*, 1), it may lead to poor representation capacity; By contrast, a larger number of heads may result in information redundancy. To

Table 5: **Ablation study of varying the number of heads in the ROI embedding layer for fMRI decoding task**. We report the classification accuracy of CSI-1 on BOLD5000 dataset. For PatchEmbed, $n = 16$ is the best choice reported in (Chen et al., 2022).

| Method | Size | | | | | |
|---|---|---|---|---|---|---|
| | 8 | 16 | 24 | 32 | 64 | 96 |
| PatchEmbed (Chen et al., 2022) | - | 26.38 | - | - | - | - |
| ROIEmbed (*Ours*) | 24.75 | 24.35 | 26.69 | **28.48** | 23.96 | 26.08 |

Table 6: **Ablation study of varying the number of heads in the ROI embedding layer for fMRI encoding task**. Here, we report Pearson correlations between the ground truth and the predicted fMRI signals of CSI-1 on BOLD5000 dataset.

| Method | Size | | | | | |
|---|---|---|---|---|---|---|
| | 8 | 16 | 24 | 32 | 64 | 96 |
| PatchEmbed (Chen et al., 2022) | - | 32.88 | - | - | - | - |
| ROIEmbed (*Ours*) | 30.72 | 30.44 | 31.82 | **34.32** | 29.67 | 31.04 |

further investigate the effect of this parameter, we report results with different values of heads in the second row of Table 5 and Table 6. As expected, it shows a trend of first rising and then falling. The best result is achieved when the number of heads is 32. Note that we do not tune it for each individual but set it to 32 consistently to show the efficacy and generalizability of our proposed layer.

**Analysis of Reliable fMRI decoding**. Ensuring the accuracy of fMRI decoding models is paramount, as they should decode brain activity rather than merely generate images without meaningful context. However, current approaches often rely on large pre-trained generative models capable of high-fidelity image generation, even when conditioned on random noise. This raises concerns about whether such high-quality images truly reflect brain activity, especially in zero-shot scenarios where the fMRI signals were not part of the training data. To assess the reliability of fMRI decoding models, we conduct two experiments:

- *Zero-shot image-fMRI-image reconstruction*: In this task, we use a novel image to estimate the fMRI signals, and then use the estimated fMRI signals to reconstruct the image. Successful image-fMRI-image reconstruction implies better semantic consistency for encoding and decoding.
- *fMRI-image with random noise as fake fMRI signals*: In this task, we randomly generate fMRI signals from a Gaussian distribution, which significantly deviates from the real fMRI signal distribution. We then decode these fake fMRI signals to generate images. The reconstructed image should contain non-semantic information, indicating non-informative fMRI signals.

It is important to note that these tasks are not intended to demonstrate the superiority of a model on new tasks but rather to assess the reliability of methods designed for visual decoding. As shown in Figure 6, LEA excels in the image-fMRI-image reconstruction task while generating unrealistic and non-informative images when conditioned on fake fMRI signals that deviate from the distribution of real fMRI signals. In contrast, MinD-Vis still generate high-fidelity and informative images when conditioned on random noise. For in-distribution sampling, both methods perform well because fMRI signals with similar distribution have already been seen in the training set. Nevertheless, for out-of-distribution sampling, it is unclear whether the noise in brain signals is really so-called noise or unknown perceptions.

Under this circumstance, methods equipped with powerful image generators still produce high-quality and meaningful images, which we think is unreliable. One possible reason is that a large-scale pre-trained image generator can handle the situation of out-of-distribution (or zero-shot) inputs. However, they are hard to evaluate because we don't know if the input fMRI signal really exists or what the ground-truth image is. On the other hand, our generated images, though seemingly meaningless, are not complete noise. The model hasn't seen these distributions but appears to combine some known semantics, like sky, lawn, and

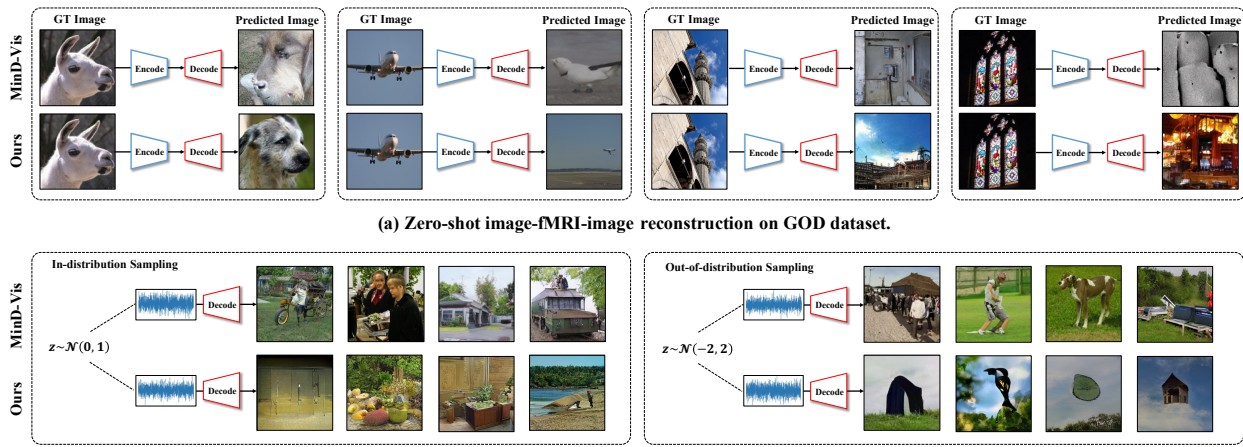

(a) Zero-shot image-fMRI-image reconstruction on GOD dataset.

(b) Neural decoding from randomly sampled noises as fake fMRI signals.

Figure 6: Reliable fMRI decoding analysis.

house. These findings underscore that existing models have not thoroughly explored the reliability of fMRI decoding models. In comparison, both the fMRI decoding and encoding components of LEA demonstrate greater reliability than current algorithms.

# 5    Broader Impact

Our proposed LEA model provides a unified framework, to efficiently recover visual stimuli from fMRI signals and predicts brain activity from images. It could serve as a tool to analyze human perception and uncover underlying cognitive mechanisms. We hope it can inspire more valuable and innovative studies in the future, and promote the development of neuroscience. However, the potential negative impact lies in that the usage of this method may cause privacy breaches, since people don't want others to pry into their thoughts. Besides, this technology shouldn't be used illegally for commercial or cognitive strategies. As a result, governments and officials may take action to govern the use of relevant datasets and technologies, and researchers should avoid using datasets that may raise ethical concerns.

# 6    Conclusions

In this paper, we address the challenge of jointly performing fMRI decoding and encoding within a single, unified framework. Our proposed Latent Embedding Alignment (LEA) not only constructs latent spaces for fMRI signals and images but also aligns them, enabling bidirectional transformation. By mitigating issues related to redundancy, instability, and data scarcity in fMRI datasets, LEA excels at producing high-fidelity, semantically consistent fMRI decoding results. Moreover, LEA demonstrates the capability to accurately estimate human brain activity from visual stimuli through fMRI encoding. Our experimental results on two benchmark datasets validate the effectiveness of LEA, highlighting its significant contribution in neuroimaging.

# 7    Acknowledgement

The computations in this research were performed using the CFFF platform of Fudan University. Yanwei Fu is with the School of Data Science, FudanISTBI—ZJNU Algorithm Centre for Brain-inspired Intelligence, Fudan University, Shanghai Key Lab of Intelligent Information Processing, and Technology Innovation Center of Calligraphy and Painting Digital Generation, Ministry of Culture and Tourism, China. Yanwei Fu is the corresponding author.

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

# A  Appendix

## A.1. Model Architectures

Our proposed Latent Embedding Alignment (LEA) framework aims to build the bridge between images and fMRI signals. To this end, we incorporate two distinct encoder-decoder architectures to learn latent representations for these two modalities by performing the self-supervised reconstruction task. Subsequently, we introduce a linear model to efficiently facilitate the transformation between the latent representation spaces of images and fMRI signals.

**fMRI Reconstruction.** We adopt the architecture of masked autoencoder (He et al., 2022) as the encoder-decoder model for fMRI signals. More concretely, the fMRI encoder consists of 24 transformer layers, each of which has 1024 feature dimensions with 16 heads. We append a [CLS] token to the input fMRI signals before feeding them into the fMRI encoder. After self-attention processing, the [CLS] token is expected to capture the global representation of the entire input fMRI signals. Next, we mask all other latent tokens and encourage the decoder to reconstruct fMRI signals from the learned [CLS] token. Similarly, the decoder includes 8 transformer layers, with 512 feature dimensions and 16 heads. Different from (Chen et al., 2022) that equally splits fMRI signals into patches, we instead maintain the voxel structure with ROI regions by additionally adding ROI-dependent processing layers (*i.e.*, ROI Embed and ROI Predict) before and after the encoder and decoder, respectively. Both ROI-dependent layers are two-layer modules with a symmetric structure and not shared among different ROI regions. Taking the ROI Embed layer as an example, it first adopts a convolution layer with 32 kernels to extract multi-head features of the input fMRI signal and then uses a fully-connected layer to project the different lengths of fMRI signal into a unified dimension of 1024, leading to the final fMRI embeddings for one ROI as $32 \times 1024$.

**Image Reconstruction.** Image-based latent space construction has been widely investigated. As our target is to decode the semantic content of fMRI signals, we adopt the pre-trained CLIP (ViT-H/14)[1] as the image encoder and the yielding CLIP latent space as our image latent space. To generate images from the image latent space, we utilize MaskGIT[2] as the image decoder in this paper, but other generative models are also acceptable. Specifically, the decoder is a decoder-only transformer layer, with a depth of 24, a feature dimension of 768, and 16 heads. By conditioned on the image latent features, the generation procedure starts from all [MASK] tokens and autoregressively predicts logits of tokens with higher confidence scores. It is a non-sequential generative procedure, following a cosine mask scheduling function.

## A.2. Data Preprocessing and Augmentation

**Preprocessing.** We use fMRI data originally collected and pre-processed from BOLD5000 (Chang et al., 2019) and GOD (Horikawa & Kamitani, 2017) datasets to conduct experimental analyses. Briefly, Chang

---

[1] https://github.com/mlfoundations/open_clip
[2] https://github.com/google-research/maskgit

et al. (2019) follow the BOLD-processing workflow of FMRIPREP[3] and extract data features via the general linear model (GLM). Horikawa & Kamitani (2017) preprocess data including 3D motion correction by SPM5[4], co-registration to anatomical images and interpolation by $3 \times 3 \times 3$ mm voxels. Please refer to the "fMRI Data Analysis" section in (Chang et al., 2019) and the "MRI data preprocessing" section in (Horikawa & Kamitani, 2017) for more details.

**Augmentation.** We utilize data augmentation to enlarge training data for both fMRI and image reconstruction and discard them all during the inference or the fitting of linear models. For fMRI signals, we only use one augmentation method by randomly adding Gaussian noise with the probability of 0.25. For image data, we apply random cropping, random horizontal flipping and finally resize images to $256 \times 256$.

### A.3. Training and Testing Procedures

All experiments are implemented with the PyTorch toolkit. The encoder-decoder model for fMRI reconstruction is initialized with pre-trained weights from HCP datasets (Chen et al., 2022). We finetune it with both fMRI training and testing sets for each individual. Note that using fMRI testing data is a common practice as in (Chen et al., 2022; Beliy et al., 2019), and we do not access any paired fMRI-image testing data when learning linear models. For image reconstruction, we use a frozen CLIP encoder and utilize the image decoder of MaskGIT initialized by ImageNet (Deng et al., 2009), and further finetune it with images from the training sets on both the GOD and BOLD5000 datasets. Once trained, it is shared for all subjects and experiments. AdamW optimizer is applied with $\beta_1 = 0.9$, $\beta_1 = 0.95$ and batch size 8 to finetune both models. The initial learning rate is defined as 5e-5 with 0.01 weight decay. We linearly decrease the learning rate until it reaches the minimal rate. The total number of training iterations for fMRI reconstruction and image reconstruction is 100k and 300k. For fitting linear models, we adopt ridge regression from the Scipy library. The coefficient of L2 regularization term $\alpha$ is set to 500 for both image-to-fMRI and fMRI-to-image regressions. During the inference of fMRI decoding, the number of steps for MaskGIT is 11 as default, and we generate 5 samples for all evaluations.

### A.4. More Qualitative Results

In this section, we show more visualizations of fMRI decoding (Figure 7 and Figure 8) and image-fMRI-image reconstruction (Figure 9).

---

[3]https://fmriprep.readthedocs.io/en/latest/workfows.html
[4]http://www.fil.ion.ucl.ac.uk/spm

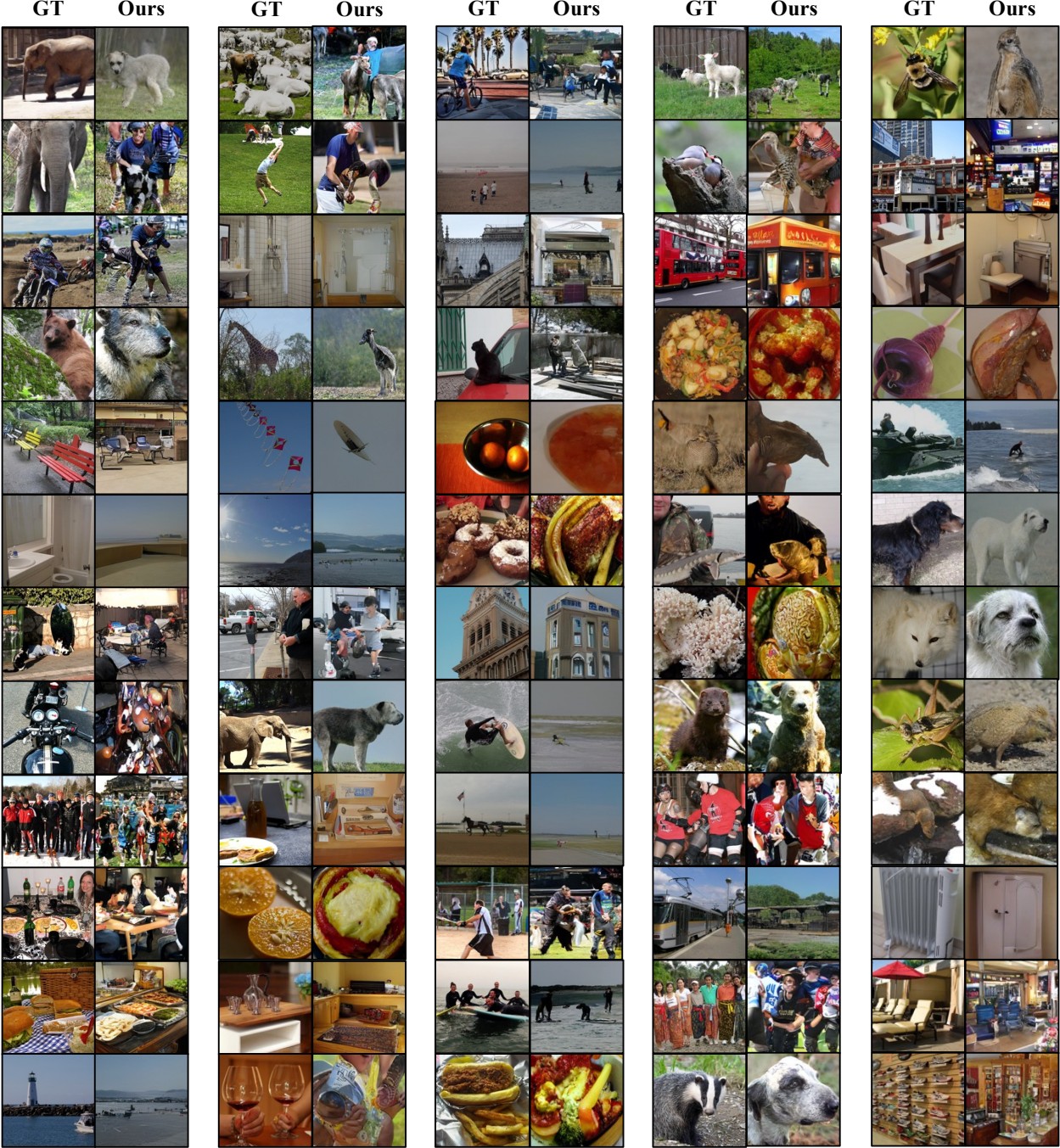

Figure 7: Samples of fMRI decoding on CSI-1 of BOLD5000 dataset.

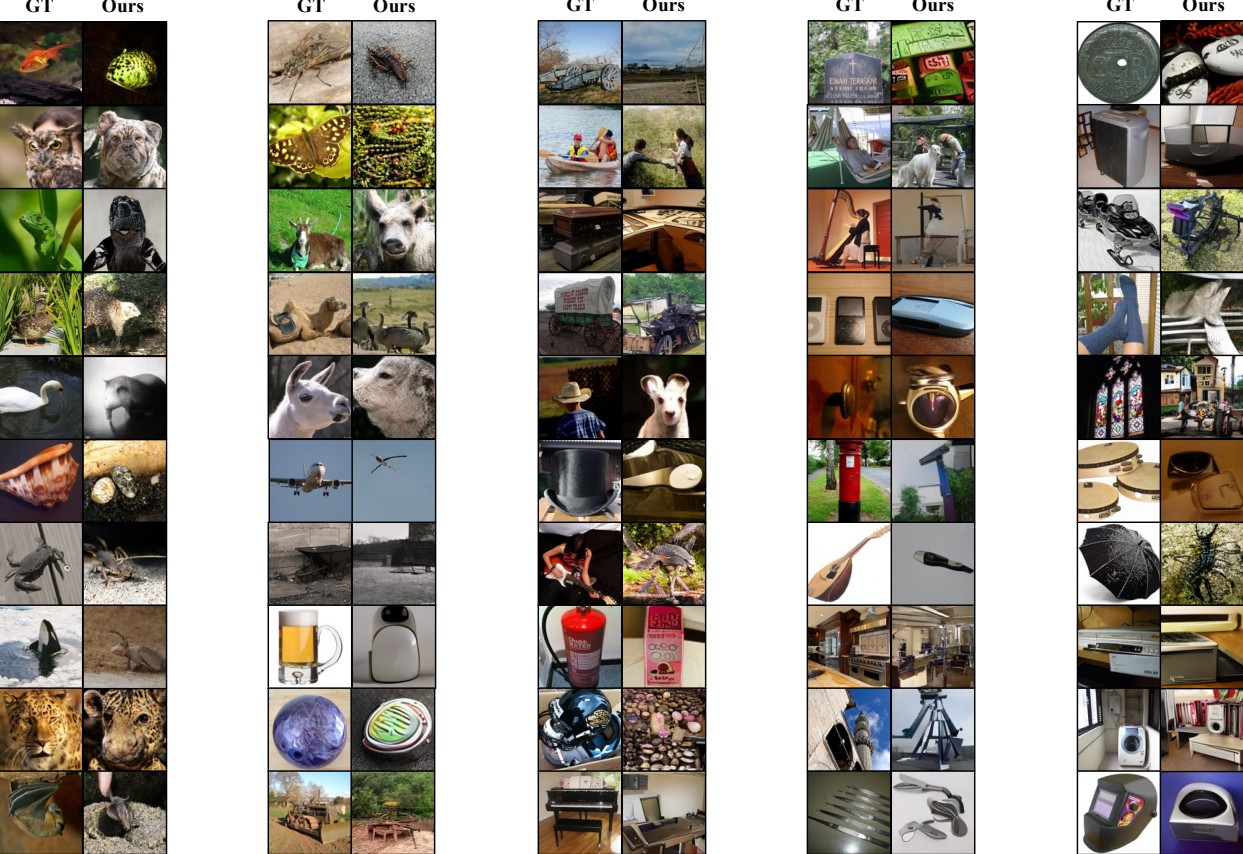

Figure 8: Samples of fMRI decoding on Sbj-3 of the GOD dataset.

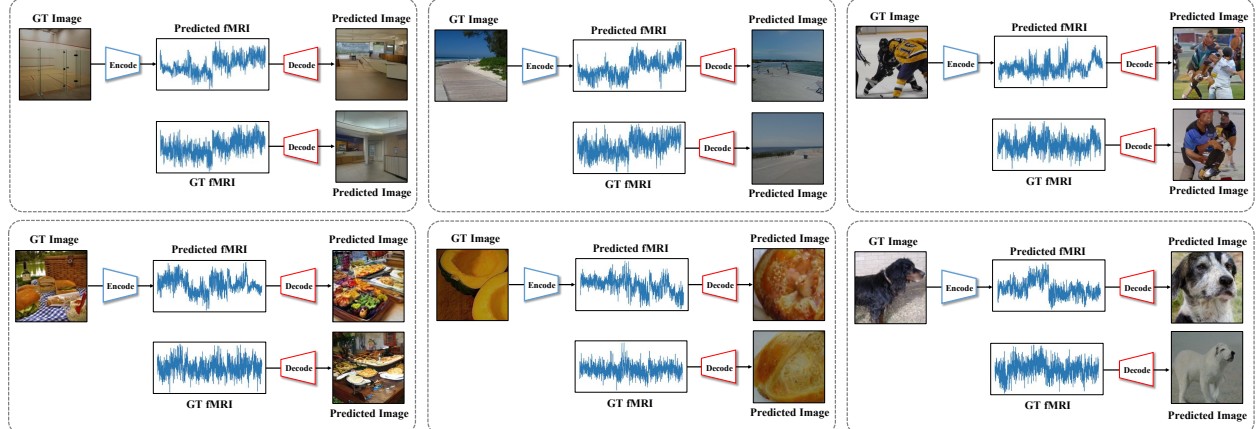

Figure 9: Visualizations of image-fMRI-image reconstruction on CSI-1 of BOLD5000 dataset.

