# OpenReview forum: "LEA: Learning Latent Embedding Alignment Model for fMRI Decoding and Encoding"
_TMLR — Accepted by TMLR_

### Review · Reviewer_KKrJ · 2024-04-14

**Summary Of Contributions:**

The paper introduces a method (LEA) for learning and aligning two latent spaces -- that of an fMRI signal, and that of corresponding visual images. The proposed method predicts bidirectionally, and can recover either fMRI signal from image, or image from fMRI signal. Of particular note is that this approach does not require (too much) paired fMRI-image data, which is a constraint in previous methods which attempt to directly apply seq2seq style approaches to learn mappings (though certainly previous methods also solve this problem in similar ways). The approach is to use CLIP embeddings for the vision latent space and to introduce the ROI embedding layer, a masked autoencoder applied to fMRI data segmented by ROI region. Once the latter model is trained from a lot of (but non-paired) fMRI data, linear maps between the latent spaces are learned on a small set of paired data, which then fill in the gap for the whole decoding/translation pipeline. The methodology is evaluated on several relevant metrics and datasets. They also add an important additional test, where they present fake noise fMRI signals to determine whether the image generation capabilities appropriately degrade.

**Audience:**

Yes

**Broader Impact Concerns:**

None.

**Claims And Evidence:**

Yes

**Requested Changes:**

1. Add comparisons to the methods mentioned above, as they are quite similar to various parts of the pipeline and are simpler. It would be good to understand if we can replace more complex methodology with simpler methodology, and what the delta is when resorting to more complex methods. It would also be good to provide better attribution for previous works (mentioned above) which outline very similar ideas to the ones presented in this paper.

2. It would be great to add a more thorough analysis of the experiments performed in Fig. 6 - perhaps some time spent coming up with a qualitative measure of this property could be fruitful, as well as being highly relevant for future works.

**Strengths And Weaknesses:**

*Strengths*:
The methods are evaluated on several datasets and metrics and generally come out ahead compared to the competitors. The paper also highlights issues with previous methods, which should generate nonsense when provided with a noise fMRI signal, but which instead generate sensible high-fidelity images, and qualitatively starts to demonstrate that their model may have better properties on this metric (generate nonsense when provided with nonsense).

*Weaknesses*:
My main hesitations about this paper derive from insufficient benchmark method comparisons and the lack of citations of earlier, highly related work. Currently, the paper lacks comparisons to earlier work in aligning multi-subject fMRI and learning alignment maps between shared fMRI representation spaces and other embedding spaces - it would be ideal to understand the improvement of the neural method proposed in this work over the simpler linear Shared Response Model (SRM) approach to learning a latent embedding space for fMRI signals across subjects (https://proceedings.neurips.cc/paper/2015/hash/b3967a0e938dc2a6340e258630febd5a-Abstract.html), as well as the resulting linear alignment maps between fMRI space and image space (https://www.sciencedirect.com/science/article/pii/S1053811917305128, where the paired representation is text). The ROI embedding layer may also benefit from a comparison to the earlier works (https://arxiv.org/abs/1608.04846, https://arxiv.org/abs/1609.09432). These works (and the first two mentioned) also contain similar considerations to the ROI embedding layer, but are a simple linear approach and another convolutional autoencoder approach, and thus natural baselines and comparisons.

Another question I had was about Figure 6 - I’m not sure I completely agree that the images generated on noise from LEA were particularly more uninformative and unrealistic (perhaps just the house in the sky?) It would be nice if there were a more thorough study of this component of the paper, since it seems like quite an important test to better understand whether the models are vacuously generating images or not. Likewise for the zero-shot task.

---

> ### Comment · Reviewer_KKrJ · 2024-09-20
> **Official Recommendation**
>
> In the current state I would vote to reject - I think the first requested change is necessary for acceptance.

---

> ### Author Response · Authors · 2024-09-21
> **Authors' Response to Reviewer KKrJ**
>
> Thanks for your time reviewing and providing professional feedback for our paper. Here are our responses to specific questions.
>
> > **Q1: Missing comparisons to some related work and the lack of citations of earlier.**
>
> Thanks for your valuable comments. (1) Firstly, we would like to explain that the Shared Response Model (SRM) approach is about aligning fMRI representations among multiple subjects, which takes as input time-series fMRI data $\mathbf{f} \in \mathbb{R}^{v \times d}$. It differs from our paper, where we focus on the technical development of within-subject fMRI decoding/encoding, and the input dimension is $v$ only.
> (2) However, these works are indeed related to our paper to some extent. For example, (Vodrahalli et al., 2018) use the linear model to learn bidirectional mappings between fMRI responses (after SRM) and natural language representations. In Tab. 2 of our manuscript, we have compared our method with a simple linear regression, which demonstrates the efficacy of our proposed fMRI encoder and decoder.
> (3) As suggested, we further conduct experiments by applying the method of (Vodrahalli et al., 2018) to our fMRI decoding task on GOD dataset. Results in the table below show that our LEA consistently achieves higher performance than (Vodrahalli et al., 2018), which suggests the superiority of our design. We will cite all relevant papers, and add the experiment and discussion in the revision.
>
>
> | **METRICS** | **METHODS** | **SBJ-1** | **SBJ-2** | **SBJ-3** | **SBJ-4** | **SBJ-5** | **AVERAGE** |
> |--|--|--|--|--|--|--|--|
> | FID | (Vodrahalli et al., 2018) | 3.61 | 2.77 | 2.34 | 2.95 | 2.55 | 2.84 |
> |  | LEA (relax) | 1.45 | 1.43 | 1.52 | 1.48 | 1.25 | **1.23** |
> |  | LEA | 2.57 | 1.51 | 1.67 | 1.71 | 1.34 | 1.76 |
> | CLIP Corr. | (Vodrahalli et al., 2018) | 69.55 | 76.38 | 79.57 | 75.34 | 75.48 | 75.26 |
> |  | LEA (relax) | 73.96 | 78.20 | 80.94 | 78.73 | 70.86 | 76.54 |
> |  | LEA | 73.79 | 79.63 | 79.96 | 76.69 | 76.32 | **77.28** |
> | CLIP Dist. | (Vodrahalli et al., 2018) | 0.35 | 0.39 | 0.41 | 0.39 | 0.40 | 0.39 |
> |  | LEA (relax) | 0.38 | 0.40 | 0.41 | 0.40 | 0.37 | 0.39 |
> |  | LEA  | 0.38 | 0.41 | 0.40 | 0.39 | 0.41 | **0.40** |
> | Acc. | (Vodrahalli et al., 2018) | 9.51 | 16.48 | 18.79 | 13.35 | 13.28 | 14.28 |
> |  | LEA (relax) | 11.18 | 18.62 | 20.45 | 20.04 | 13.15 | **16.69** |
> |  | LEA | 11.35 | 18.55 | 18.92 | 17.52 | 14.45 | 16.16 |
>
>
> > **Q2: More explanations about the study in Fig. 6.**
>
> Thanks for the question. We first clarify that the study in Fig. 6 is not intended to show the superiority of our model on some new tasks, or to conclude who is good or bad. All methods are designed for visual decoding, and we analyze their reliability with the help of image-fMRI-image reconstruction and visual decoding from random noises. Here, `reliability' refers to our thought that a powerful image generator may make the decoding ability of the model flashy. Concretely,
>
> (1) In Fig. 6(a), a good image-fMRI-image reconstruction means better semantic consistency for encoding and decoding. Our predicted image may not be the same as the input image, but they still share some consistent semantics.
>
> (2) In Fig. 6(b), for in-distribution sampling, all methods perform well because fMRI signals with similar distribution have already been seen in the training set. For out-of-distribution sampling, the core is what the noise means in brain signals, is it really so-called noise or unknown perceptions? We don’t know.
>
> Under this circumstance, other methods equipped with powerful image generators still produce high-quality and meaningful images, which we think is unreliable.
> One possible reason is that a large-scale pre-trained image generator can handle the situation of out-of-distribution (or zero-shot) inputs. However, they are hard to evaluate because we don't know if the input fMRI signal really exists or what the ground-truth image is.
> On the other hand, our generated images, though seemingly meaningless, are not complete noise. The model hasn't seen these distributions but appears to combine some known semantics, like sky, lawn, and house.
>
> Last but not least, more visualizations of out-of-distribution sampling are uploaded online with an anonymous link (https://drive.google.com/file/d/1GE6OE5zX1prLJOASrqNR7MlrLdahOVf4/view?usp=sharing) for review.

---

> > ### Comment · Reviewer_KKrJ · 2024-09-27
> > **Thanks for the update!**
> >
> > Re: Q1, thanks for running the comparisons, this is great! My concern here is resolved.
> > Re: Q2, I strongly agree with the sentiment "powerful image generators still produce high-quality and meaningful images, which we think is unreliable" - I was only suggesting it would be an even stronger point if there was a more thorough evaluation developed, but it is completely reasonable that it is outside the scope of this paper. I would point to the development of a more reliable metric for assessing the extent of the 'reliability' in this context in future work.
> >
> > With these changes, I think the paper is reasonable to accept.

---

> > > ### Author Response · Authors · 2024-10-07
> > > **Thank you**
> > >
> > > We sincerely appreciate your effort in helping us to strengthen the paper and your support for our work! We will punctiliously revise the paper based on the above discussion and experimental results, which can definitely make our paper more solid.

---

### Review · Reviewer_rbVG · 2024-06-12

**Summary Of Contributions:**

The authors present a method for fMRI encoding (predicting fMRI activations given a visual stimulus) and decoding (reconstructing the stimulus image given the fMRI activations).
To do so, they use 2 separate encoder-decoder pairs, one for (stimulus) images and one for fMRI signals.
Then they train linear models to learn the mappings from the latent space used to embed stimuli to the latent space for fMRI signals and vice-versa.
Training the encoder-decoder architectures separately for fMRI and natural images, and using a simple linear model to link the 2, allows exploiting available fMRI-only or image-only data, and requiring only a small amount of aligned (fMRI, image) pairs which are scarce.

The authors assess the encoding and decoding capabilities of their approach on the BOLD and GOD datasets, which contain an important amount of fMRI data on a small number of subjects.
They compare their method to several baselines from the literature, which the new method outperforms.

**Audience:**

Yes

**Broader Impact Concerns:**

The "broader impact" statement feels overly cautious -- there is a gap from decoding the broad category of image someone is looking at with low accuracy after hours of fMRI recording and "reading people's mind".

**Claims And Evidence:**

Yes

**Requested Changes:**

Could the authors provide more details on the "fMRI signal f" -- its dimensions, how it was obtained from the BOLD timeseries, etc. In the N x L matrix, IIUC N is the number of images and L the number of voxels that are in one of the considered ROIs. What are the values of N and L? It might be interesting to display a brain map of the correlations, so that we can see which regions' activity is well predicted by the encoding model.

I am not sure I understand exactly which testing data is used and for which task in the LEA("relax") setting. Is the encoding model allowed to see the test stimuli (but not the corresponding fMRI data) during training?
Why is LEA(relax) present on table 3 but not table 1; and does "ours" refer to LEA or LEA(relax)?

"Empirical findings suggest that a similar linear relationship exists between fMRI- and vision-only models" -- could the authors provide a reference?

"on various benchmark datasets" -- here IIUC "various" means 2 (BOLD5000 and GOD)?

Do the authors intend to share their code and trained models online after publication?

**Strengths And Weaknesses:**

Strengths: the authors propose a method that outperforms the baselines both for encoding and for decoding. The method for embedding and decoding images, fMRI signals and for linking the 2 latent spaces is well described.

Weaknesses: some aspects of the experimental setting could be clarified (see below).

---

> ### Author Response · Authors · 2024-09-21
> **Authors' Response to Reviewer rbVG**
>
> Thanks for your time reviewing and providing professional feedback for our manuscript. Here are our responses to specific questions.
>
> > **Q1: Could the authors provide more details on the 'fMRI signal f'.**
>
> Thanks for the question and we apologize for any confusion. Let us respond to your question one by one.
>  * $\mathbf{f}$ is a (1-D) vector fMRI signal. This format is widely used by previous approaches (Ozcelik et al., 2022; Ferrante et al., 2022; Chen et al., 2022; Ozcelik & VanRullen, 2023). The dimension of $\mathbf{f}$ varies across different subjects due to individual differences, thus, we omit the definition of its dimensions in Sec. 3.1, but provide a sample process of RoI Embed layer in Sec. A.1.
>  * $N$ is the number of testing images or fMRI signals. And $L$ is the dimension of fMRI signals, which varies across different subjects. For example, four subjects on BOLD5000 have a signal length of 1685/1129/1466/2787.
>  * fMRI data is collected and pre-processed as the way in their original papers (Chang et al., 2019) and (Horikawa & Kamitani, 2017). Briefly, the fMRI data is measured by asking subjects to look at a natural image over a while. Then, a time series of fMRI data is entered into general linear model (GLM) to produce the so-called fMRI signal $\mathbf{f}$.
>
>
> > **Q2: Which testing data is used and for which task in the LEA("relax") setting?**
>
>   * Assuming we train a model on BOLD5000 dataset, for the 'relax' setting, the corresponding fMRI testing data is additionally used (not the corresponding test stimulus) for training the reconstruction task via a self-supervised manner. When learning the latent embedding alignment module, we do NOT access any paired fMRI-image testing data, and the parameters of both fMRI and image reconstruction modules are frozen.
>   * In Tab. 4, 'Ours' refers to 'LEA (relax)'. We sincerely apologize for the clerical error and will correct it in the revision.
>   * Comparisons between with and without 'relax' are more like an ablation study to discuss whether using the test fMRI data (not reveal labels) for self-supervised learning may lead to information leakage. So, we just report their results in one table each for fMRI encoding and decoding tasks to allay this concern.
>
> > **Q3: Could the authors provide a reference to support the statement that 'Empirical findings suggest that a similar linear relationship exists between fMRI- and vision-only models'.**
>
> Thanks for the question. We would like to explain that 'empirical findings' here refer to our pilot study to support our motivation derived from (Merullo et al., 2023). We will make it clearer in the revision.
>
>
> > **Q4: "on various benchmark datasets" means 2?**
>
> Yes, we conducted experiments of three tasks on both BOLD5000 and GOD datasets for evaluation. We will tone down the writing to make it more precise.
>
> > **Q5: Do the authors intend to share their code and trained models online?**
>
> Definitely! Our codes and models will be released after publication.

---

### Review · Reviewer_PUR8 · 2024-09-04

**Summary Of Contributions:**

This paper presents an approach for predicting the stimulus image presenting from fMRI data (fMRI decoding) and predicting the fMRI signal given the visual stimuli (fMRI encoding). The approach first trains separate auto encoding reconstruction models for fMRI signal and images separately, then learns a linear transformation to map the latent embeddings of paired fMRI and image data. The method is tested and compared against baselines on multiple fMRI-visual stimulus datasets.

**Audience:**

Yes

**Broader Impact Concerns:**

A broader impact statement section is included, and while it discusses some possible concerns (though many are still far-reaching considering the current state of the results), I think it could be helpful to also discuss the potential effects of using the approach when the results are inaccurate/incorrect (eg the fMRI decoding prediction is wrong)

**Claims And Evidence:**

No

**Requested Changes:**

1. It is unclear what exactly the input is the fMRI data f. While image I is given dimensions, f is not defined. Reading the ROI embedding section, it seems that the fMRI may be the BOLD image data from a single time point, but the imaging is dynamic, so how is the frame chosen? For pre-training this is less a concern, in that each frame could be used for learning the reconstruction (although the dynamic information is then lost), but for decoding the fmri signal -> image stimuli and vice versus, how was the correspondence between fMRI frame and visual stimulus determined?

2. The paper mentions "standard pre-processing" for fMRI, but there is no real "standard". At least a brief description of the type of fMRI preprocessing performed is needed.

3. The variable M is used to represent 2 quantities (number of ROIs and total amount of training data) - different variable names should be used to avoid confusion.

4. The paper states that the image encoder comes from a CLIP pretrained model, but there are no details regarding what dataset is used to train the CLIP encoder. Please clarify.

5. To learn the latent fMRI embeddings, the authors state they take an "individual-specific approach", where the embedding is learned per subject. I am confused then how generalizable a model would be, both in terms of machine learning standpoint of applying to new subjects and new data, but also in a practical sense, since if the model needs to be tuned for each subject, you would always need to have some paired data available for each new subject, which is not necessarily feasible. Could the authors please discuss and address these concerns.

6. The metrics used to evaluate the results need more explanation. The authors use what they call "CLIP correlation" and "CLIP distance". I am still not sure what the CLIP correlation is based on the description in the paper, and the CLIP distance seems to actually be a similarity measure (where higher is better) which is confusing, and also I don't know how exactly the similarity is measured. Please edit and clarify.

7. The authors note that the fMRI testing data is used for learning the reconstruction task in self-supervised manner. While it is true that no "label" is used, this still means that the learned embedding space will be based on the testing data, and thus is not really an independent test dataset. Please provide some comment / explanation if this is the case.

8. Some of the methods have "relax" as part of the model name. Although the authors note that some setting corresponds to relax, I don't see in contrast what the corresponding method without relax refers to. Could the authors please clarify both what it means to have relax and not have relax appended to the model name.

9. In Fig. 4, qualitative examples are shown from Ozcelik et al., but I do not see any corresponding quantitative results that would be expected to be shown in Table 3. The results from this comparison should be included.

10. The compared PatchEmbed method is only tested on one size (n = 16), which actually outperforms many RoIEmbed settings that tested other dimensions for the proposed approach. It would be good to see the effect of different size patches for PatchEmbed to show that RoIEmbed really gives an advantage.

11. Overall, the paper still needs a lot of editing for polishing language for clarity.

**Strengths And Weaknesses:**

Strengths:
- The problem motivation is of interest in neuroscience - understanding the relationship between visual stimuli and brain signals.
- The proposed approach, which first learns the reconstruction models then aligns the latent embeddings with a linear model, allows for training the individual domain auto encoding models from larger amounts of data followed by learning from smaller amount of paired data to link the two latent domains.
- Multiple experiments with different evaluations are performed to assess the proposed method.
- The approach nominally performs as well or better than many of the compared baselines.

Weaknesses:
- Many method details are unclear or missing, which are key to understanding the approach
- Some experimental methods details are unclear or missing, which are key to evaluating the results
- Some additional experiments would help provide more evidence for some model choices
- The results of the paper, while showing some overall improvement, are based on a small number of subjects (5 from one dataset, 4 from another). I recognize this is in part due to the availability of such a dataset, but it is hard then to say whether the results show true improvement over baselines, and further, whether the results would be generalizable.
- The paper needs additional proofreading and editing for language and typos.

Please see next section for specific details/questions regarding the weaknesses

---

> ### Author Response · Authors · 2024-09-21
> **Authors' Response to Reviewer PUR8**
>
> Thanks for your time reviewing and providing professional feedback for our manuscript. Here are our responses to specific questions.
>
> > **Q1: What the input is the fMRI data $f$, and how is the correspondence between fMRI frame and visual stimulus determined?**
>
> Thanks. We would like to politely clarify some misunderstandings below.
>   * *fMRI frame and visual stimulus are pre-processed by their original papers:* Briefly, the fMRI data is measured by asking subjects to look at a natural image over a while. Then, a time series of fMRI data is entered into general linear model (GLM) to produce the so-called fMRI signal $\mathbf{f}$. Therefore, the correspondence between fMRI data and visual stimulus is determined by the way of collection and pre-processing in their original papers (Chang et al., 2019) and (Horikawa & Kamitani, 2017), which is beyond the scope of this paper.
>   * *$\mathbf{f}$ is a (1-D) vector represented fMRI signals:* As stated in the paragraph of 'ROI embedding layer', $\mathbf{f}$ is a 1-D vector and this format is widely used by previous approaches (Ozcelik et al., 2022; Ferrante et al., 2022; Chen et al., 2022; Ozcelik & VanRullen, 2023). Since its length varies across different subjects due to individual differences, we omit the definition of its dimensions in Sec.3.1, but provide a sample process of RoI Embed layer in Sec. A.1.
>
> > **Q2: What is 'standard pre-processing'?**
>
> We apologize for the confusion. 'standard pre-processing' in the paragraph of 'ROI embedding layer' means the pre-processing used in the original papers of datasets. Please refer to the 'fMRI Data Analysis' section in (Chang et al., 2019) for BOLD5000 dataset and the the 'MRI data preprocessing' section in (Horikawa & Kamitani, 2017) for GOD dataset. We will revise the writing and add citations.
>
> > **Q3: The variable M is used to represent 2 quantities.**
>
> Thanks for pointing it out and apologize for the clerical error. We will make changes in the revision.
>
> > **Q4: Please clarify the dataset used for CLIP pre-training.**
>
> Thanks. We adopt CLIP ViT-H/14 pre-trained on LAION-2B. The details have been discussed in Sec. A.2.
>
> > **Q5: 'individual-specific approach' is not generalizable for new subjects or data.**
>
> Thanks for raising the question. Please allow us to explain this point from three aspects below.
>   * *'Cross-subject evaluation' is extremely challenging:* Cross-subject evaluation is a challenge to the whole sub-community of this topic, even extremely hard tasks for fMRI research papers. Lots of related studies also mainly focus on individual-specific evaluation (Ozcelik et al., 2022; Ferrante et al., 2022; Chen et al., 2022; Liu et al., 2023; Du et al., 2023; Ozcelik & VanRullen, 2023). One of the reasons is that the fMRI signal undergoes substantial influence from individual differences. For example, four subjects on BOLD5000 have a signal length of 1685/1129/1466/2787.
>   * *The positioning of our paper is the same as previous related work:* Similar to previous work, our paper focuses on the technical development of within-subject fMRI decoding/encoding, which has been clearly stated in Sec. 3. The study of cross-subject evaluation or generalization is beyond the scope of this paper.
>   * *Our work enjoys the advantage of efficient alignment:* Simplicity is one of the advantages of our proposed latent embedded alignment module, which allows this module to learn from scratch within a few minutes. Instead, previous approaches like (Chen et al., 2022) need to fine-tune or retrain the model for different subjects.
>
> > **Q6: The metrics of 'CLIP correlation' and 'CLIP distance' need more explanation.**
>
> Thanks. These two metrics are based on CLIP features between generated images and ground-truth images. Concretely, 'CLIP correlation' is the 2-way comparison of CLIP features via pair-wise correlation, and 'CLIP distance' means the similarity of two CLIP features via cosine distance.
> We will make it clear in the revision and our codes will be public.

---

> ### Author Response · Authors · 2024-09-21
> **Authors' Response to Reviewer PUR8 (Cont.)**
>
> > **Q7: Please provide some comment/explanation about the usage of fMRI testing data for learning the reconstruction task.**
>
> Thanks for bringing up this discussion.
> We totally understand this concern. Strictly speaking, test sets indeed cannot be used for model training. However, the number of fMRI signals on GOD and BOLD5000 is very limited. Ideally, the more data the fMRI reconstruction module is trained on, the better the feature representation, and thus the better the latent embedding alignment between fMRI features and image features. So for a fair comparison, we follow the previous work (Chen
> et al., 2022; Beliy et al., 2019) and use the fMRI test set without revealing the label information. Specifically, the fMRI testing data is only trained for the reconstruction task via a self-supervised manner. When learning the latent embedding alignment module, we do NOT access any paired fMRI-image testing data, and the parameters of both fMRI and image reconstruction modules are frozen.
>
> In addition, we also provide experiments on GOD dataset for both fMRI encoding and visual decoding tasks, by training the model without using any fMRI testing data (as stated in the paragraph of 'Setting' in Sec. 4.1). Results in Tab. 2 and 3 show that both settings get comparable performance, which indicates that the learned embedding space is not fully dominated by the testing data
>
>
> > **Q8: Please clarify what it means to have 'relax' or not appended to the model name.**
>
> Thanks. As we explained in the paragraph of 'Setting' in Sec.4.1, the model with the suffix 'relax' means that fMRI testing set is used for the self-supervised learning of the fMRI reconstruction task. Otherwise, we do not use any testing data in all training stages, which is distinguished by the model name without the suffix 'relax'.
>
> > **Q9: The results of Ozcelik et al. should be included in Tab. 3.**
>
> Thanks for the valuable comment. As suggested, we calculate the metrics of generated images from (Ozcelik et al., 2022). The results below show that our proposed method outperforms it across all metrics, especially for FID and accuracy, which is echoed by the quantitative visualizations in Fig. 4. We will add the experiments and discussions in the revision.
>
>
> | **METRICS** | **METHODS** | **SBJ-1** | **SBJ-2** | **SBJ-3** | **SBJ-4** | **SBJ-5** | **AVERAGE** |
> |--|--|--|--|--|--|--|--|
> | FID | ICGAN (Ozcelik et al., 2022) | 5.22 | 3.72 | 3.65 | 4.42 | 3.69 | 4.14 |
> |  | LEA (relax) | 1.45 | 1.43 | 1.52 | 1.48 | 1.25 | **1.23** |
> |  | LEA | 2.57 | 1.51 | 1.67 | 1.71 | 1.34 | 1.76 |
> | CLIP Corr. | ICGAN (Ozcelik et al., 2022) | 74.02 | 80.83 | 77.65 | 75.44 | 76.46 | 76.88 |
> |  | LEA (relax) | 73.96 | 78.20 | 80.94 | 78.73 | 70.86 | 76.54 |
> |  | LEA | 73.79 | 79.63 | 79.96 | 76.69 | 76.32 | **77.28** |
> | CLIP Dist. | ICGAN (Ozcelik et al., 2022) | 0.39 | 0.40 | 0.42 | 0.39 | 0.41 | **0.40** |
> |  | LEA (relax) | 0.38 | 0.40 | 0.41 | 0.40 | 0.37 | 0.39 |
> |  | LEA  | 0.38 | 0.41 | 0.40 | 0.39 | 0.41 | **0.40** |
> | Acc. | ICGAN (Ozcelik et al., 2022) | 10.71 | 15.95 | 18.79 | 14.52 | 12.31 | 14.46 |
> |  | LEA (relax) | 11.18 | 18.62 | 20.45 | 20.04 | 13.15 | **16.69** |
> |  | LEA | 11.35 | 18.55 | 18.92 | 17.52 | 14.45 | 16.16 |
>
>
> > **Q10: It would be good to see the effect of different size patches for PatchEmbed.**
>
> Thanks. In our pilot study, we have evaluated the effect of $n=8$ and $n=32$ for PatchEmbed. Both get inferior results to $n=16$. Since we did not run other experiments, we only reported the results of n=16 in Tab. 5 and 6, which is also the best choice for (Chen et al., 2022). As discussed in the first subsection of Sec. 4.4, it is reasonable that our RoIEmbed achieves better results at larger sizes, since a small number of sizes for each region after RoIEmbed may lead to poor representation capacity. Note that we do not tune it for each individual but set it to 32 consistently. We also beat (Chen et al., 2022) in Tab. 2~4, showing the efficacy of our design.

---

### Decision · Action_Editor_NE4T · 2024-11-13

**Recommendation:** Accept with minor revision

**Comment:**

The consensus is that the paper is OK in general, but does not include enough explanation on the fMRI preprocessing procedures. These explanation shave to be included.

Besides, I found several typos. I list a few below, but the text obviously needs help from a grammar checker / speller.

* Abstract: "it is still challenge" ->  "it is still challenging"
* "Our approach starts by avoiding the need" does not sound right
* " the issue of signal redundancy present in fMRI data" -> does not make sense. redundacy brings robustness for encoding/decoding...
* "assessing the reliability of both fMRI encoding and decoding models through the image-fMRI-image reconstruction assessment." assessing/assessment
* p.2 "when predicting brain activities" -> "when predicting brain activity"
* p.3 bottom, misplaced comma
* "the trained decoder can subsequently used" -> "the trained decoder can subsequently be used"
* RoI or ROI ?
* section 3.4 is too informal, and should be rewritten more thoroughly
* "the fMRI testing data is only trained for the reconstruction task via a self-supervised manner." -> "the fMRI testing data is only trained for the reconstruction task in a self-supervised manner."
* table 2 presents percents I guess ?
* "With the absent of" -> "With the absence of"
* "Human brains are known to spatially separated, that different RoI regions..." -> "Human brains are known to spatially segregated, that different RoIs..."
* "number of head" -> "number of heads"
* "On the contrast" -> "by contrast"
* "to efficiently recovers" -> "to efficiently recover"
* "read people’s mind." is too colloquial and too simplistic. Please rephrase
* "its significant contributions in the field." -> "its significant contribution in neuroimaging."
* "decnoder" -> "decoder"

**Audience:**

This paper is for audience interested i cognitive neuroscience.

**Claims And Evidence:**

The Claims of the paper are clearly justified by both qualitative and quantitative experiments.
Quantitative experiments involve encoding and decoding task performed on two datasets. They include several baselines.
Ablation studies are also presented.

The main issue outlines by all reviewers is the lack of explanation of preprocessing done on fMRI data, in particular how an fMRI sample is computed for each trial and input image.